# Improved Visual SLAM Using Semantic Segmentation and Layout Estimation

**Ahmed Mahmoud *** and **Mohamed Atia**

Department of Systems and Computer Engineering, Carleton University, Ottawa, ON K1S 5B6, Canada
* Correspondence: ahmedgmahmoud@cmail.carleton.ca

**Abstract:** The technological advances in computational systems have enabled very complex computer vision and machine learning approaches to perform efficiently and accurately. These new approaches can be considered a new set of tools to reshape the visual SLAM solutions. We present an investigation of the latest neuroscientific research that explains how the human brain can accurately navigate and map unknown environments. The accuracy suggests that human navigation is not affected by traditional visual odometry drifts resulting from tracking visual features. It utilises the geometrical structures of the surrounding objects within the navigated space. The identified objects and space geometrical shapes anchor the estimated space representation and mitigate the overall drift. Inspired by the human brain's navigation techniques, this paper presents our efforts to incorporate two machine learning techniques into a VSLAM solution: semantic segmentation and layout estimation to imitate human abilities to map new environments. The proposed system benefits from the geometrical relations between the corner points of the cuboid environments to improve the accuracy of trajectory estimation. Moreover, the implemented SLAM solution semantically groups the map points and then tracks each group independently to limit the system drift. The implemented solution yielded higher trajectory accuracy and immunity to large pure rotations.

**Keywords:** visual SLAM; layout estimation; semantic SLAM; visual navigation





## 1. Introduction

As a human or an animal navigates through an unfamiliar environment, some form of spatial memory is formed. This memory creates a cognitive map representing a spatial environment and contains information about metric and directional relationships of objects in that environment. Humans are highly visual creatures [1] and, under typical situations, vision forms a critical foundation for representing space [2]. While extensive research on navigational behaviour and spatial memory has been carried out in rodents, memory research in humans has traditionally focused on more abstract, language-based tasks [1].

Recent studies have begun to address the gap in human navigation research benefiting from the virtual reality evolution that enabled virtual navigation simulations combined with human electrophysiological recordings [3]. These studies suggest that the medial temporal lobe [4] (MTL) is equipped with a population of place and grid cells similar to that previously observed in the rodent brain. We believe that recent neuroscience discoveries related to cognitive memory could revolutionise VSLAM solutions. In this section, we discuss the neuroscientific answers to the following questions:

- Does the human brain perform SLAM?
- How does the human brain solve navigation tasks?
- What landmarks and semantic information could form human brain spatial maps?
- What is the brain spatial mapping process?

John O'Keefe, Moser, and Edvard were awarded the Nobel Prise for medicine in 2014. They won the prise for discovering two types of cells in the hippocampus: place cells and grid cells. These findings significantly impacted cognitive neuroscience, particularly spatial

cognition. Further research found other cell types in the hippocampus, boundary vector cells, and head direction cells [5–7].

Place cells are activated when an animal is in a specific position in an environment, regardless of the aspects of the job at hand [8]; they include data about a specific area. Grid cells fire at regular intervals and represent the layout of an environment. When an animal is near a given edge of an environment, boundary vector cells show activity, providing additional information about the animal's relative position in its surroundings. The head direction cells signal the animal's head orientation. Combining these cells that code for position, layout, borders, and head orientation creates a mental map of the environment in a useful state for navigation. This mental map depicts the spatial organisation of an environment without regard to a particular point of view.

Finally, it is essential to acknowledge the brain motor sensor that gives the brain an initial guess of the body motion that helps the brain control the eye movements to prevent disorientation and always anticipate where surrounding objects should be [9,10]. In addition, our brain optimises the memory and calculations needed to perform its navigation tasks by abstracting the environment to the space boundaries and objects referenced to these boundaries [11–14].

### 1.1. Human Navigation Strategies

We use three basic ways to plan our trajectory to reach our goal: allocentric, egocentric, and beacon [15]. The allocentric or spatial memory strategy is a navigational strategy that involves intricate geometric calculations [16]. These calculations take distance and directional information into account [17]. An environmental representation (map) is formed and referenced outside one's current body position. The navigator explores the environment by establishing relationships between various landmarks and orienting oneself with respect to those landmarks [18]. Creating external maps is among the human navigation skills [19–21]. Still, they are helped by a variety of more complicated cognitive activities that extract abstract semantic information from landmarks [22], routes taken, and the environment's architecture [23–26].

Recent work in human spatial navigation has shown that the environmental boundaries provide a way to establish an allocentric coordinate system [27,28]. The surrounding spatial geometry, such as the square or rectangle shape defined by an environment's boundaries, can be a powerful cue for organising externally referenced knowledge. Several investigations [29–32] have reported that aligning the objects with the environment axis increases the accuracy of the brain's awareness of the locations of the objects.

Egocentric representation [32] is a strategy that is more typically utilised in everyday scenarios such as reaching for a pen or remembering where the key chain is in the room. Our current body position is the reference in egocentric representations. The pen is on the table in front of us, about 30 degrees around our current facing orientation. We frequently use this type of representation to avoid collisions with objects and traverse our immediate, peripersonal space, as demonstrated by various studies of human spatial cognition [33–35].

Several studies imply that egocentric representations are high-resolution visual snapshots linked to our current head direction. We may create a single coherent egocentric representation related to our current location in space by taking a succession of these high-resolution, static, body-referenced images [36]. These representations can then be updated as we walk through an environment, constituting the basis for path integration [22,37]. However, these representations diminish during disorientation [34–36] or in large-scale environments [38].

The third navigation strategy is a beacon or reaction method [8,39–41]. It requires learning a series of behavioural actions from specific environment points that act as stimuli. With a succession of stimulus-response associations, one can learn to navigate from home to work in an automated manner. These responses may involve turning at a specific corner or building, with the corner and building acting as stimuli and the response involving turn-

ing [42–44]. The reaction approach is inflexible when the remembered route is unavailable since it cannot create a novel way to the target place.

As a result, navigation tasks are as simple as moving closer to or farther away from a specific object. Its position on the retina grows or shrinks, giving an important clue for locating the object. When combined with egocentric codes like "right" and "left", beacon navigation is a replica of how we travel using mobile devices like GPS on our phones. Like GPS, beacon strategies decrease the navigator's job to simply search for a specific landmark and link it with a response.

### 1.2. Mapping and Navigation Tasks in the Human Brain

We presented neuroscience and behavioural analysis findings describing how our brain solves navigation tasks and forms a surroundings model. These findings could be summarised as follows:

- The human brain builds short-term, high-resolution body-referenced visual maps (egocentric) to be used with the motor sensors' motion predictions to solve the immediate navigational tasks; however, these maps fade with time and when navigating in large spaces (environmental spaces). These maps are the equivalent of the local maps that VO and VSLAM systems use in pose estimation.
- Alongside the egocentric maps, the human brain forms a more general representation that integrates visual and motor sensors over time, generating allocentric environmental models. The allocentric maps are referenced outside the human body to a distinct point or a landmark in the environment. This type of model is equivalent to the global maps in SLAM systems.
- Unlike SLAM systems in the literature, the human brain depends on the environment's geometry and layout to recognise and map the visited places.
- The brain mainly tracks the surrounding objects for pose estimation. It does not depend on feature points unless they are essential for describing the space layout or belong to a distinct landmark. Alternatively, the brain abstracts all the background points into colour and shade information.

These navigational guidelines resulted from generations of training and evolution of the human brain, the most sophisticated computer that has ever existed. Designing a SLAM solution based on these findings could yield scalable, more robust, optimised solutions.

## 2. Related Work

This section discusses the related research in layout estimation and semantic SLAM applications.

### 2.1. Layout Estimation

Layout estimation research considers the task of estimating the spatial layout of an indoor scene from single or multiple images. Layout estimation solutions aim to delineate the walls, ground, and ceiling boundaries [44]. Early layout estimation solutions approached the problem intuitively as a semantic segmentation problem, assigning geometric context classes (floor, walls, and ceiling) to each pixel and then trying to obtain room layout keypoints and boundaries based on the pixel-wised labels [45].

However, it is nontrivial to identify layout keypoints and boundaries from the raw pixel output. Furthermore, the traditional pixel-based representation created ambiguity as researchers have shown that CNNs often have difficulty distinguishing between different surface identities [46]. This phenomenon largely undermines the overall room layout estimation performance.

RoomNet [47] reformulated the task of room layout estimation as an ordered room layout keypoint localisation problem, which can be directly addressed using CNNs as a trainable end-to-end problem. RoomNet considered a list of different layout possibilities with their respective keypoint definitions that could be inferred from a single image; these possibilities are first introduced by [48] and are listed in Figure 1. These 11 layouts cover

most possible situations under typical camera poses and common cuboid representations under the "Manhattan world assumption" [49]. Once the trained model estimates correct keypoint locations with an associated room type, we can simply connect these points in a specific order to produce a boxy room layout representation.

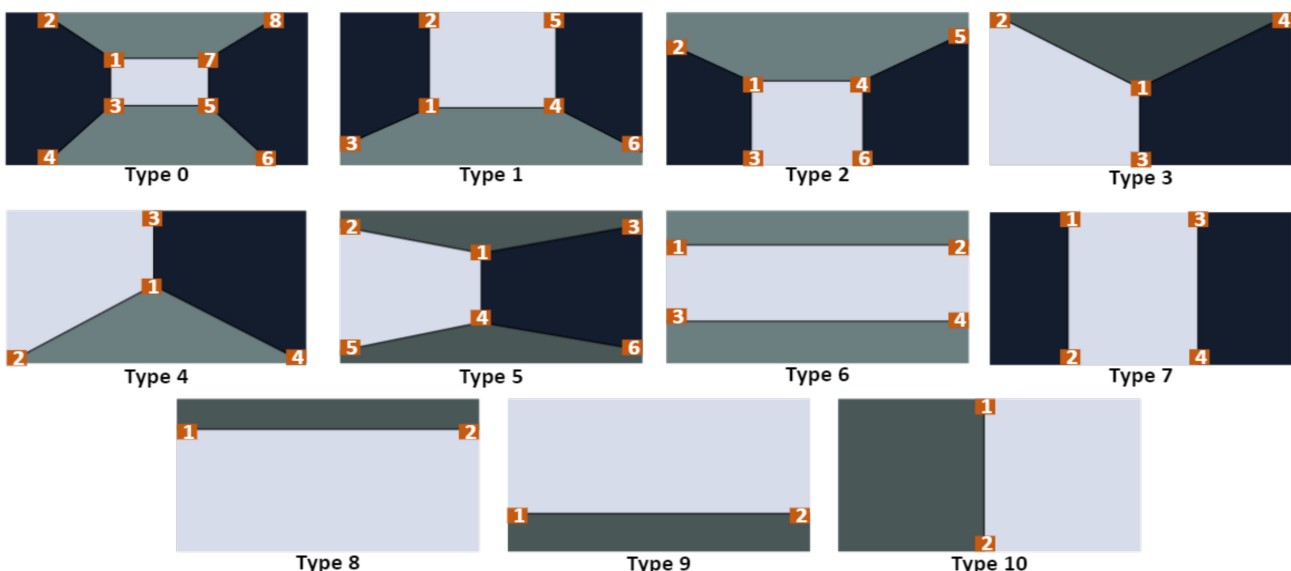

**Figure 1.** Definition of scene layout types.

RoomNet adopts the SegNet architecture proposed in [50] with modifications. The base architecture of RoomNet adopts the same convolutional encoder–decoder network as SegNet. Using an image of an indoor scene, the system can directly identify a set of 2D room layout keypoints to recover the room layout. Each keypoint is represented by a 2D Gaussian heatmap centred at the true keypoint location. The encoder–decoder architecture processes the information flow through bottleneck layers, enforcing it to implicitly model the relationship among the keypoints that encode the 2D structure of the room layout.

RoomNet predicts room layout keypoints and the associated room type with respect to the input image in one forward pass. To achieve this goal, the channels in the output layer are 11 as the number of the layout scenarios shown in Figure 1. The side head selects the output channel with fully connected layers to the bottleneck [51–54], as shown in Figure 2.

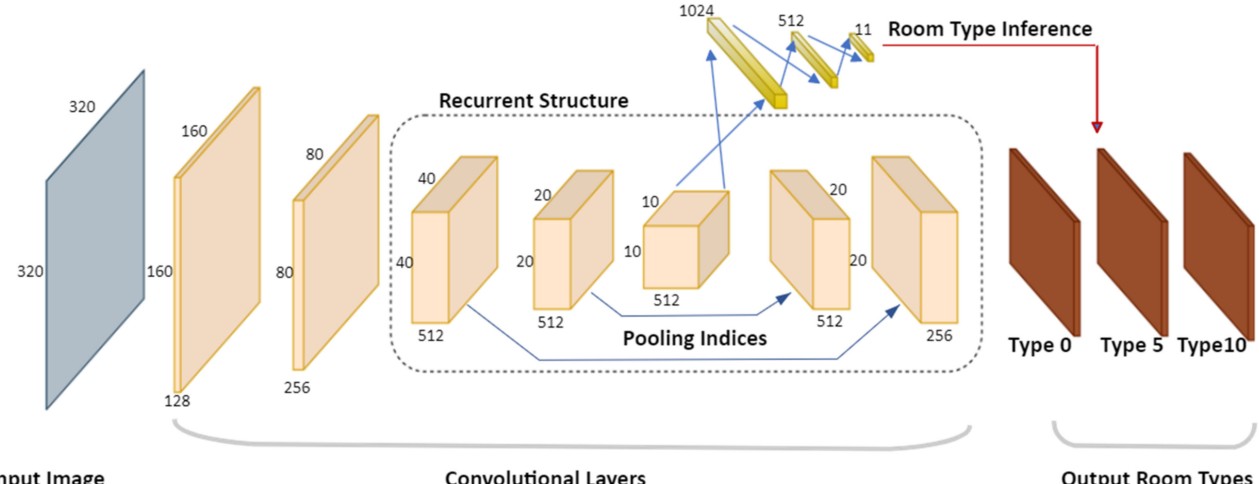

**Figure 2.** RoomNet base architecture.

## 2.2. Semantic SLAM

Purely geometry-based maps could not provide conceptual knowledge of the surroundings to facilitate complex tasks; as a result, associating semantic concepts with geometry entities in the environment has recently become a popular research domain. Semantic SLAM incorporates semantic information into the SLAM process to improve overall performance by providing task-driven perception, resilience, and high-level understanding. Incorporating semantic data into a SLAM pipeline has been around for a long time.

Nonetheless, the implementation was not achievable until recently, at least in a meaningful fashion, because extracting semantic information from visual data was nearly impossible until the epoch of deep learning. When fast and accurate semantic segmentation DNN architectures existed, the semantic SLAM drew much attention. However, because this occurred less than a decade ago and, despite several contributions, the topic of integrating semantics in SLAM is still far from mature [55]. This section identifies and summarises the primary topics of interest in semantic SLAM solutions.

### 2.2.1. Semantics for Loop Closing

Since it is challenging to detect a loop closing purely based on geometric features, robust loop closing is an open problem in SLAM. Changes in the observed environment, such as moving objects or changes in illumination, can cause dramatic changes in the geometry of a scene. When semantics are used, the problem becomes much easier to solve.

Some researchers have concentrated their efforts on mathematically formulating the problem of combining geometric and semantic information into a single optimisation framework for semantically associating observations and performing robust loop closing [56]. Others used semantic information to train a neural network to extract 3D descriptors of the scene [57]. To detect loop closures, 3D descriptors are compared rather than words in a bag of binary words setting [58].

### 2.2.2. Semantics for Handling Dynamic Environments

Traditional SLAM pipelines have a critical flaw: they operate under the assumption that the environment is static. The RANSAC algorithm is commonly used in SLAM to eliminate erroneous feature correspondences. However, when a significant portion of the view is occupied by moving objects this approach fails, causing the estimation process to diverge. Semantics can be used in various ways to deal with this situation. Semantic segmentation is used to detect potentially moving parts of the image and exclude them from the tracking process entirely in a simple but effective approach. Mask-SLAM [59], which uses DeepLabv2 [60] to perform precise semantic scene segmentation, is one approach. Mask-SLAM discards ORB features detected within regions occupied by vehicles or the sky because vehicles move and the sky is too far away to show any parallax.

Before discarding potentially moving objects found using semantic segmentation, more complicated methods check if they move. DS-SLAM [61], for example, examines whether the matched features are close to the epipolar line between frames. If multiple matched features belonging to a potentially moving object are far from the epipolar line, the object is considered moving. Li et al. [62] proposed a more complicated approach to avoid moving objects when estimating camera motion, but instead of discarding them, try to track them independently through time. They use an off-the-shelf deep network to get 2D bounding boxes of vehicles, then train their own CNN to create 3D bounding boxes based on the 2D ones. Based on this information, they can then track each moving vehicle separately using a motion model and an object bundle adjustment to the 3D bounding boxes camera static.

SaD-SLAM [63] proposed an RGB-D feature-based SLAM system that utilises semantic masks obtained using Mask_RCNN [64]. SaD-SLAM does not discard the entire features of the moving objects; instead, it uses epipolar constraints to identify the static features from dynamic objects to improve the accuracy and robustness of the solution. AirDOS [65] proposed a novel approach to benefit from the fixed geometrical relations between feature

points on the rigid moving objects to enhance the camera pose estimation accuracy. Like SaD-SLAM, AirDOS benefits from the geometrical relations on the rigid body and builds a 4D spatio-temporal map that includes both dynamic and static objects. AirDOS identifies the potential moving objects using instance segmentation. Then, it uses the static features to estimate the initial ego-motion of the camera. For moving objects, the feature points are triangulated and then their positions are tracked using optical flow. Finally, the system removes the erroneous points before performing BA based on rigid body constraints.

2.2.3. Semantic Reasoning within an Unknown Environment

Semantic reasoning appears to be the most researched among the methods listed above. Since this is probably the simplest way of integrating semantics in SLAM, much research has created a dense or sparse semantic map of the environment. In this case, the SLAM framework assigns a semantic class to each map point produced by a semantic segmentation method. Even though the semantic maps generated by such methods can enable advanced interactions between the system and the outside world, most do not use them to improve the SLAM's robustness.

Building semantic maps with objects rather than 3D points is a unique approach. Object detectors have been used in place of feature detectors in QuadricSLAM [66,67]. The detected objects are inserted into the map in a dual quadric representation. The dual quadrics positions and the camera pose are then estimated using a bundle adjustment. Because ignoring traditional features reduces SLAM's accuracy, Hosseinzadeh et al. [68] proposed integrating dual quadric representations of objects and traditional 3D points. Instead of abstract object representations, a more accurate but computationally expensive and complicated solution integrates detailed volumetric object reconstructions in the semantic map. By fusing the geometry of objects in successive planes, the object reconstructions are gradually refined over time. Similar concepts are implemented in a few published works, such as MaskFusion [69] and Fusion++ [70].

A more straightforward concept has been developed by Wang et al. [71]. As a starting point, ORB-SLAM2 [72] is employed. Then, the YOLO object detector [73] is used to semantically classify the frame's keypoints on each frame. A voting system then propagates semantic information from keypoints to corresponding map points. All keypoints associated with the map point "vote" once for their semantic class. After integrating semantic information, semantics are used in feature matching, tracking, and loop closing. More specifically, associations between keypoints and map points of different classes are prohibited or reduced based on an adjustable ratio.

## 3. Proposed System

We propose a semantically improved visual SLAM solution inspired by human reasoning in solving navigational tasks. The proposed SLAM builds an accurate joint map for sparse feature models for foreground objects, the environment's geometric bounds and detected sparse feature points to represent the background. The system depends on a RoomNet trained network to estimate the layout keypoints in the input image sequences. RoomNet considers a cuboid shape of the indoor spaces and infers its cuboid corners. The proposed system then incorporates the layout constraints into the global optimisation process to enhance the overall trajectory accuracy.

The proposed system uses ORBSLAM2 as its SLAM backbone and propagates the semantic and geometric inferences across the tracking, mapping, and loop closure tasks with the following contributions:

1.  A complete visual SLAM system tracks and maps geometric and semantic structures in indoor environments. The system was built on ORB-SLAM2 as its SLAM backbone and redesigned its threads to utilise semantic observations throughout the tracking, mapping and loop closing tasks.

2.  The system also uses an offline created object model database to provide a point of view independent object tracking and provide the physical relationship between the object points as a set of constraints that anchor the odometry positional drifts.
3.  Because background feature points are hard to precisely re-detect and match, the proposed system is configured to allow object points to be more influential in pose optimisations.
4.  With most of the images in the tracked sequences containing at least one cuboid corner, tracking and mapping threads are modified to detect and track cuboid corners and include them in the map optimisation process.
5.  The proposed system introduces the idea of slicing the map into multiple geometrical links. Each link represents keyframes connecting two environment corners and exploits the geometric relationship between these corners as map optimisation constraints.
6.  Two modifications are introduced to the loop closure thread: first, the local map is queried for objects and cuboid points to verify loop detections before accepting it. The second modification introduces a new loop closure approach by detecting all the cuboid corners of the traversed space. Four cuboid corners allow the system to optimise the four edges defining the space limits.

The proposed system was tested on multiple datasets generated from the AI2Thor environment simulator-embodied AI agents [74]; more details on the test data are presented in the Experiments and Results section.

## 4. System Description

Figure 3 shows an overview of the proposed system. Like ORB-SLAM2, the developed system consists of three threads: tracking, local mapping, and loop closing. Each thread was redesigned to exploit the semantic and geometric information to accurately estimate the trajectory and environment map. The system embedded an offline learnt object model database for the common objects in the AI2Thor [74] scenes. This section presents a detailed description of the system's threads, highlighting the modified (grey) and the new (orange) components introduced to the ORB-SLAM2.

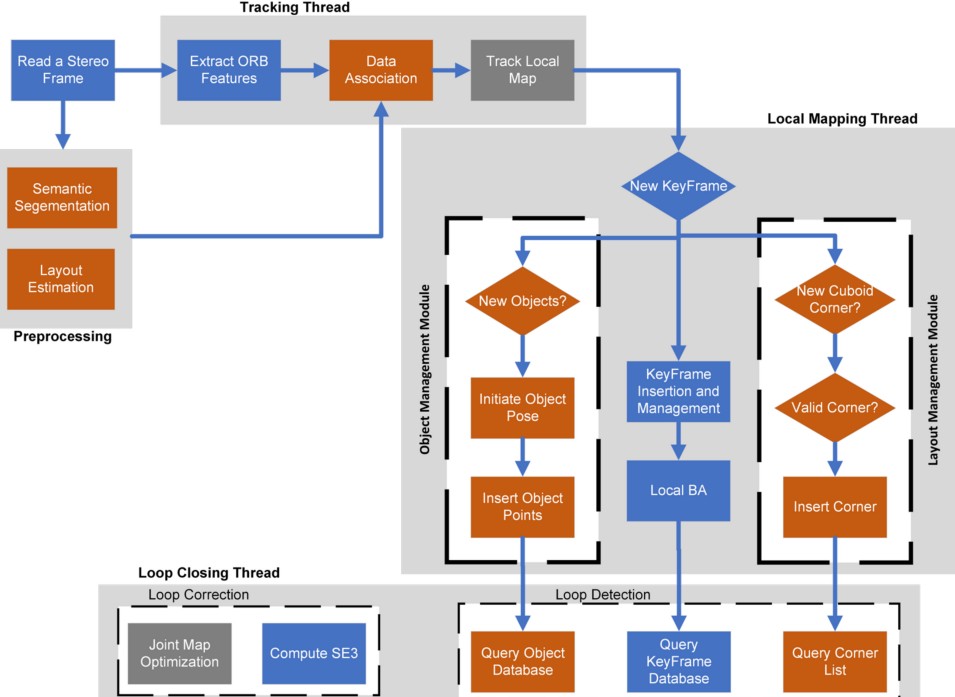

**Figure 3.** Proposed solution flow diagram.

### 4.1. Object Model Database

One of the system contributions is creating an offline sparse 3D point model database for common objects in AI2Thor simulator scenes. The proposed SLAM system loads the object model database in its initialisation and uses the modelled objects to increase the robustness and accuracy of the solution. These models act as the brain memory, which does not need to view a familiar object from all angles to recognise and track it. Additionally, these models are optional to the system. If an object was detected and did not have a model in the database, the system would still track its newly discovered features. In other words, a model could be considered a partial map that only contains a specific object's map points and is injected into the SLAM local map once the object is detected.

Figure 4 shows the model database creation and query flow diagram. A process similar to the ORB-SLAM2 mapping was used to create an object model. Twelve images of the object are taken 30 degrees apart from the same distance. Then ORB feature points are extracted and matched using the same ORB extractor and matcher from ORB-SLAM2. The matched points are then triangulated to create a point cloud of the features. For each point, the model saves the BRIEF descriptor and the transformation $T_{op}$ from the center point of the point cloud. Finally, each point also contains the mean viewing angle at which it could be detected.

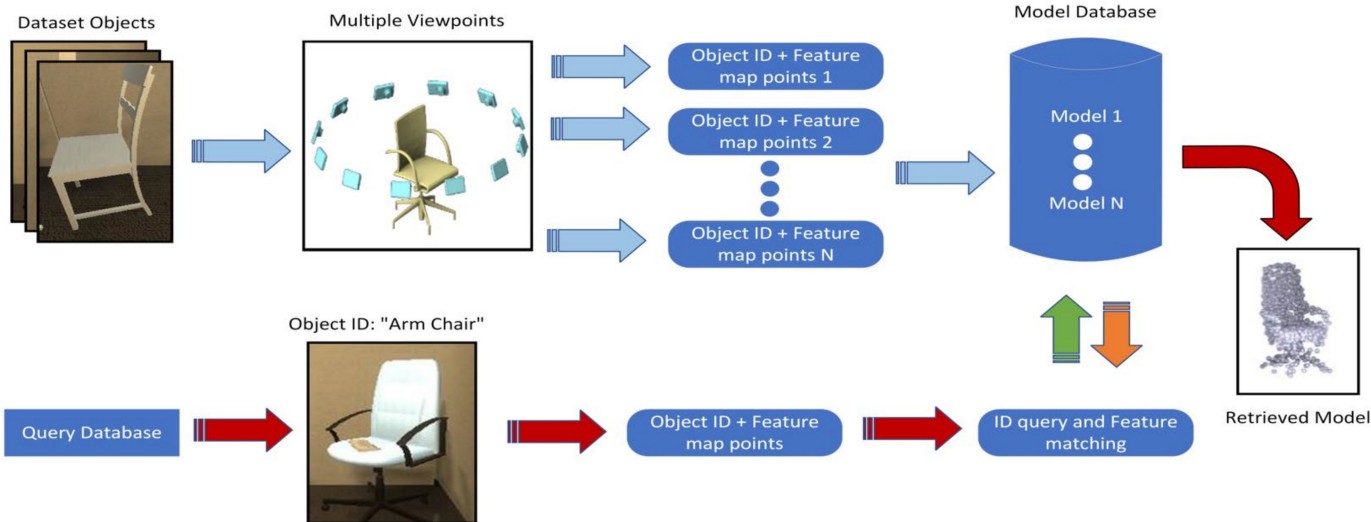

**Figure 4.** Object model database flow diagram.

To query the database for new object detection, the system uses the object ID provided by the semantic segmentation node to retrieve all models of the same kind (chair, sofa, etc.) and then perform feature matching on the retrieved models to select the right model and a rough estimate of its relative pose with respect to the camera frame. The feature matching process considers the relative pose of the features and whether the detected object is cropped by the image pounds.

### 4.2. Frame Preprocessing

While AI2Thor provides a semantic segmentation solution for each frame, the layout estimation solution is still needed. The solution should be able to identify and track the cuboid vertices of the surrounding space. RoomNet was utilised to perform this task for its compatible and efficient representation. We trained the RoomNet network on an AI2Thor-generated dataset. The room type and layout keypoint indices are generated for the trajectory sequences. Finally, our presented system only considers room types 0–5, as types 6–10 provide no real room corners (refer to Figure 1).

*4.3. System Threads*

4.3.1. Tracking Thread

The tracking thread localises the camera with every input frame by finding feature matches to the local map and minimising the reprojection error by applying motion-only BA. The flow of the proposed system's tracking thread is as follows:

1. After extracting ORB features from the current frame, all detected features are annotated using the semantic segmentation input data.
2. The previous frame is searched for matches; the encoded semantic annotations guide the search, limiting the mismatches and the number of match candidates. An initial estimate of the pose is then optimised using the found correspondences.
3. Once an initial pose is acquired and we have an initial set of matches, we project the local map points into the current frame; this includes observed feature points, object points, and layout corner points. If the point projection lies inside the image bounds and with a viewing angle less than $60°$, we search the still unmatched features in the current frame and associate the map point to the best match.
4. Finally, the camera pose is optimised with motion-only bundle adjustment, a variant of the generic BA described in Chapter 2, where the camera $c$ pose in the world frame $w$ ($\boldsymbol{T}_{cw} = [\boldsymbol{R}_{cw}|\boldsymbol{P}_{cw}]$) is optimised to minimise the reprojection error between the 3D map points $\boldsymbol{X}_w$ matched to the current frame's 2D keypoints $x_c$:

$$\boldsymbol{T}_{cw} = \underset{\boldsymbol{R}_{cw},\boldsymbol{P}_{cw}}{\mathrm{argmin}} \sum_j \rho \left( \| x_c^j - \pi_m \left( \boldsymbol{R}_{cw} \boldsymbol{X}_w^j + \boldsymbol{P}_{cw} \right) \|_{\Sigma_j}^2 \right). \tag{1}$$

All map points included in the pose optimisation are fixed; only the camera pose is optimised. Our system's optimisation process is modified to respect the physical relationship governing the map points belonging to the same object. The system considers each matched map point an observation of either an object or the background. Background points are usually map points that lie on surfaces like floors, walls, and ceilings. These surfaces are generally homogeneous, making it hard to track background points accurately; yet background points are usually the majority of the tracked map points. We propose performing the pose optimisation with multiple subsets of the matched points. Each subset is composed of the map points belonging to the same object instance to remedy the effect of erroneous background matches. The final estimate is the mean of the subset estimates as shown in the equation:

$$\boldsymbol{T}_{cw} = \frac{1}{N} \sum_N \underset{\boldsymbol{R}_{cw},\boldsymbol{P}_{cw}}{\mathrm{argmin}} \sum_j \rho \left( \| x_c^j - \pi_m \left( \boldsymbol{R}_{cw} \boldsymbol{X}_w^j + \boldsymbol{P}_{cw} \right) \|_{\Sigma_j}^2 \right), \tag{2}$$

where $N$ is the number of tracked object instances; our experiments on the generated AI2Thor dataset showed that tracking objects improve trajectory accuracy.

4.3.2. Local Mapping Thread

In ORB-SLAM2, the local mapping thread manages and optimises the local map by performing local BA. The map management includes adding new keyframes and removing redundant keyframes and map points. The proposed system added object management and map layout management tasks to the local mapping thread.

Object management module: each object detection in newly created keyframes that does not exist in the local map is considered a new object. The object model database is queried for the detected object model. New objects are then aligned by computing the object center point's position and orientation using the matched points. After performing map optimisation (global/local), the objects within the optimised map are realigned. After object alignment, the local mapping creates object points for all the object model points regardless of their existence in the view of the keyframe. All points are then added to the object instance.

Layout management module: indoor environments are characterised by limited size, which enforces low-speed movements and many pure rotations to correct directions. In our collected and tested datasets (real and simulated) for cuboid spaces, we noticed that over 80% of the images contain at least one cuboid corner. In the remaining 20%, the images are either occluded by close objects or walls, suggesting that the following motion would be a large rotation to correct the direction and avoid a collision. Large pure rotations are usually miscalculated or could even cause the tracking thread to lose track. Tracking cuboid corners can restore lost tracking and correct wrongly estimated rotations. Detected corners are kept in an ordered stack in the same order of corner observations. The corner stack can only contain four corners at max. The layout management module shown in Figure 3 as a part of the local mapping thread limits the solution drifts as follows:

Each corner instance should contain a list of all object instances in the local map when the corner was observed with the relative transformation between the object instance and the corner point. Object instance lists are used in corner matching and are updated with each new corner observation.

Each cuboid corner detected in newly created keyframes that does not have matches in the local map is required to pass two tests. First, the corner should be cross-examined with the previously detected corners geometrically and through object list validation to ensure it is a new corner. Then the distance $d_{c_i,c_m}$ between the detected corner $c_i$ and the center point of the map $c_m$, which is updated with each new keyframe, should meet the following condition:

$$d_{c_i,c_m} = d_{mean} \pm 10\%, \tag{3}$$

where $d_{mean}$ is the mean distance between all the corners in the corner stack.

The first and second cuboid corners are directly inserted into the corner list. Now, the system can correct every newly detected corner by projecting the distance between the newly detected corner and the last inserted corner in the corner list in the direction of the perpendicular unit basis of the last link.

With third and fourth corner detection, the estimated map is optimised using the corrected corner point position in edge optimisations. Edge optimisation is the process in which the geometric constraints between the different corners are enforced. The optimisations are performed whether the corner is new or already included in the list as long as it is its first appearance on the current local map. An edge $e_i$ is all keyframes that connect two subsequent corners. In other words, all keyframes inserted in the map from the beginning of the observation of the corner $i$ to the end of the observation of corner $i + 1$. Edge optimisation is similar to local map optimisation with one key difference: corner points are fixed.

### 4.3.3. Loop Closure Thread

The loop closing thread is assigned to detect large loops, verify their correctness, and correct the accumulated drift by performing a pose-graph optimisation. Originally, the ORBSLAM2 detects if a loop is closed by recognising the visual bag of words of a previously visited scene. Then try to find a transformation between the two loop sides that fulfill the loop closure.

Accepting a wrong loop could have a destructive effect on the trajectory and map estimations and could yield unusable solutions. To prevent false loop associations, the proposed system added an additional verification step.

Before accepting a loop candidate, the list of objects and cuboid corners in the vicinity (local map) of the two keyframes representing the loop sides are required to find the common detected objects and cuboid corners. The loop candidate is rejected if no common objects/cuboid corners are found. It is also rejected if the relative transformation from the objects/cuboid corners to the loop sides is inconsistent.

The proposed system implemented another loop closure detection approach. The new approach is based on layout corner detection and is performed by the layout management module in the local mapping when detecting four different and valid cuboid corners. With

four corners already in the corner's list and the oldest corner is redetected, a full loop has already been navigated. In this case, we perform four edge optimisations, which is equivalent to a loop closure.

As presented in the Analysis section, this process should yield a more accurate trajectory shape and scale. In addition, edge optimisation gives a much more accurate initial state to the full graph bundle adjustment, which decreases the number of iterations needed to find the final state. Finally, the effect of adding a wrong cuboid point to the corner's list is as destructive to the solution as a wrong loop closure, which signifies the importance of the cuboid corners' acceptance tests performed by the layout management module.

## 5. Experiments and Analysis

### 5.1. Test Data

The developed system was tested on multiple sequences generated from the AI2Thor environment simulator-embodied AI agents [74]. AI2Thor was created by the Allen Institute for AI in 2016 to facilitate research in many domains, including deep reinforcement learning, object detection and segmentation, motion planning, and visual SLAM. AI2-THOR provides near photo-realistic 3D indoor scenes, where AI agents can navigate in the scenes and interact with objects to perform tasks. The simulator offers RGB and depth images, annotated semantic segmentation, and navigational ground truth [75].

The experiments were performed on five stereo/RGBD image sequences generated from the AI2Thor simulator. The sequences represent a full loop in the scenes with the following names: TrainFloor10_1, TrainFloor10_2, TrainFloor10_3, TrainFloor10_4, and TrainFloor10_5. The trajectories experienced many large pure rotations (with no combined translation) to test the robustness and accuracy of the systems. The length of the sequences varies from 1.3 K to 3.6 K steps per sequence.

AI2Thor allows data generation using simple python scripts. Data generation starts with initialising the environment and choosing the desired scene and the moving agent mode. The AI2Thor API allows the agent to interact with the environment by performing different actions such as MovingAhead, RotateLeft, RotateRight, MoveBack, Crouch, and Stand. Following each completed action, the user can export different information about the agent, onboard sensors, and the scene. The exported data includes ground truth position, colour images, depth images, semantically segmented images, and in-scene objects with their bounding boxes.

The agent, by default, has a single camera and to simulate the stereo camera, a third-party camera was added with a baseline displacement from the agent's default camera. After each action, the third-party camera position and orientation were updated to preserve the camera's stereo relation. Finally, the third-party camera images and data were exported after its position update. Figure 5 shows a sample of the exported images from the agent onboard camera.

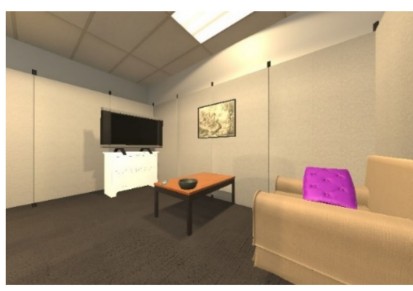
(**a**) RGB image

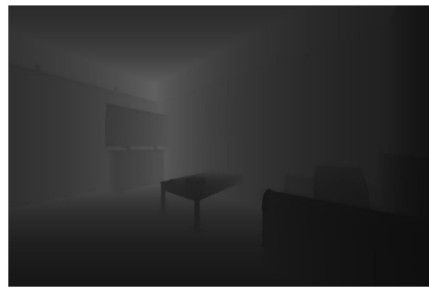
(**b**) Depth image

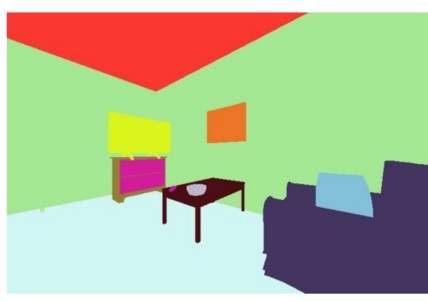
(**c**) Semantic Segmentation Image

**Figure 5.** Sample sensor data.

### 5.2. Evaluation Metrics

To evaluate the proposed SLAM system against ORB-SLAM2, the estimated camera motion for both systems is compared against the true trajectory using two frequently utilised methods: the relative pose error (RPE) and the absolute trajectory error (ATE). RPE measures the difference between the estimated motion and the true motion. It can evaluate the drift of a visual odometry system [36], which is especially useful if only sparse and relative relations are available as ground truth.

Instead of judging the drift of relative poses, the ATE tests global trajectory consistency. ATE aligns the two trajectories and then directly evaluates the absolute pose differences. This method is well-suited for assessing visual SLAM systems [34,39], but requires that absolute ground truth poses are available. Furthermore, the frame rate of the ORB-SLAM2 and the proposed system in both stereo and RGBD cases are presented to evaluate the real-time performance.

### 5.3. Results and Analysis

The quantitative comparison results are shown in Tables 1–3. RMSE, mean error, median error, and standard deviation (SD) values are presented, while RMSE and SD are more concerned because they can better indicate the robustness and stability of the system. We also show the values of improvement compared to the original ORB-SLAM2. The improvement values in the tables are calculated as follows:

$$\eta = \frac{o - b}{o} \times 100\%, \tag{4}$$

where $\eta$ represents the value of improvement, $o$ represents the value of ORB-SLAM2, and $b$ represents the value of the proposed system.

**Table 1.** Results of Metric Rotational Drift (RPE).

| Sequences | ORB-SLAM2 | | | | Proposed | | | | Improvements | | | |
|---|---|---|---|---|---|---|---|---|---|---|---|---|
| | RMSE | MEAN | Median | S.D. | RMSE | MEAN | Median | S.D. | RMSE | MEAN | Median | S.D. |
| **Floor10_1** | 4.299 | 3.472 | 3.662 | 2.537 | 2.567 | 2.086 | 2.208 | 1.496 | 40.29% | 39.92% | 39.71% | 41.01% |
| **Floor10_2** | 1.501 | 1.246 | 1.367 | 0.839 | 0.753 | 0.609 | 0.630 | 0.443 | 49.86% | 51.11% | 53.89% | 47.21% |
| **Floor10_3** | 1.395 | 0.936 | 0.728 | 1.035 | 0.798 | 0.648 | 0.639 | 0.467 | 75.88% | 24.94% | 53.66% | 62.66% |
| **Floor10_4** | 3.604 | 2.570 | 1.493 | 2.528 | 2.107 | 1.533 | 0.979 | 1.446 | 41.55% | 40.37% | 34.41% | 42.79% |
| **Floor10_5** | 3.699 | 3.231 | 3.092 | 1.799 | 2.431 | 2.126 | 2.128 | 1.179 | 34.27% | 34.20% | 31.19% | 34.48% |

**Table 2.** Results of Metric Translational Drift (RPE).

| Sequences | ORB-SLAM2 | | | | Proposed | | | | Improvements | | | |
|---|---|---|---|---|---|---|---|---|---|---|---|---|
| | RMSE | MEAN | Median | S.D. | RMSE | MEAN | Median | S.D. | RMSE | MEAN | Median | S.D. |
| **Floor10_1** | 0.0036 | 0.095 | 0.064 | 0.164 | 0.0004 | 0.032 | 0.020 | 0.058 | 87.77% | 68.55% | 66.28% | 64.62% |
| **Floor10_2** | 0.0035 | 0.153 | 0.144 | 0.109 | 0.0009 | 0.063 | 0.038 | 0.072 | 73.79% | 58.52% | 73.42% | 33.74% |
| **Floor10_3** | 0.0014 | 0.058 | 0.035 | 0.103 | 0.0003 | 0.043 | 0.016 | 0.038 | 75.88% | 24.94% | 53.66% | 62.66% |
| **Floor10_4** | 0.0038 | 0.105 | 0.076 | 0.165 | 0.0016 | 0.098 | 0.075 | 0.083 | 56.89% | 7.193% | 1.594% | 49.48% |
| **Floor10_5** | 0.0004 | 0.045 | 0.030 | 0.047 | 0.0003 | 0.041 | 0.012 | 0.045 | 11.26% | 7.752% | 60.60% | 4.074% |

**Table 3.** Results of Metric Absolute Trajectory Error (ATE).

| Sequences | ORB-SLAM2 | | | | Proposed | | | | Improvements | | | |
|---|---|---|---|---|---|---|---|---|---|---|---|---|
| | RMSE | MEAN | Median | S.D. | RMSE | MEAN | Median | S.D. | RMSE | MEAN | Median | S.D. |
| **Floor10_1** | 0.696 | 0.552 | 0.504 | 0.424 | 0.23 | 0.181 | 0.158 | 0.142 | 66.96% | 67.25% | 68.66% | 66.47% |
| **Floor10_2** | 0.367 | 0.334 | 0.353 | 0.152 | 0.15 | 0.138 | 0.145 | 0.061 | 58.97% | 58.74% | 58.92% | 60.08% |
| **Floor10_3** | 1.170 | 0.188 | 0.157 | 1.156 | 0.36 | 0.013 | 0.079 | 0.359 | 69.36% | 92.96% | 49.72% | 68.98% |
| **Floor10_4** | 0.752 | 0.655 | 0.643 | 0.369 | 0.15 | 0.132 | 0.130 | 0.075 | 79.83% | 79.85% | 79.79% | 79.76% |
| **Floor10_5** | 0.717 | 0.514 | 0.478 | 0.499 | 0.261 | 0.176 | 0.162 | 0.193 | 63.53% | 65.77% | 66.11% | 61.29% |

As we can see from Tables 1–3, by tracking objects and geometrical structures in indoor environments, the proposed SLAM solution outperforms ORB-SLAM2 by an order of magnitude. The RMSE and SD improvement values can reach 79% in terms of ATE. The results indicate that the proposed approach can significantly improve the robustness and stability of the SLAM system in indoor environments.

Figures 6–10 show the estimated trajectory of the proposed SLAM system (shown in Figures 6b, 7b, 8b, 9b and 10b) and ORB-SLAM2 (shown in Figures 6a, 7a, 8a, 9a and 10a). The tested solution was plotted against the ground truth in all presented figures. We can notice how starting the sequence by large pure rotation severely affects the ORB-SLAM2 trajectory estimation in sequences 3 and 4, which also suffered large inaccuracy in the first sequence because of multiple large pure rotations. Our solution was recovered from these rotations by tracking the environment boundaries in all cases.

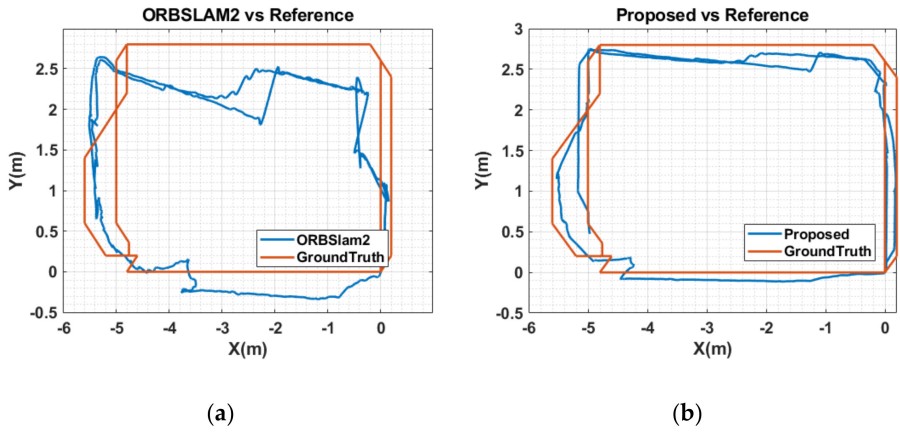

(**a**)　　　　　　　　　　　　　　(**b**)

**Figure 6.** Estimated Trajectories for Floor10_1 Sequence: (**a**) ORB-SLAM2, (**b**) SLAM system.

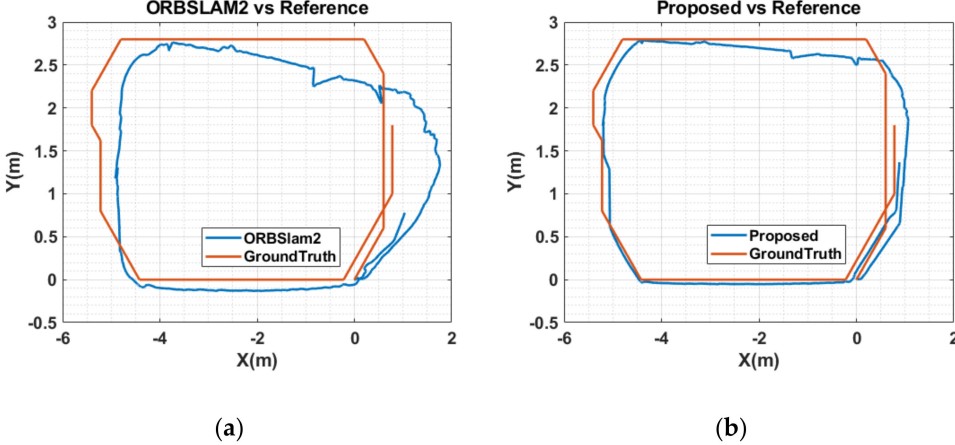

(**a**)　　　　　　　　　　　　　　(**b**)

**Figure 7.** Estimated Trajectories for Floor10_2 Sequence: (**a**) ORB-SLAM2, (**b**) SLAM system.

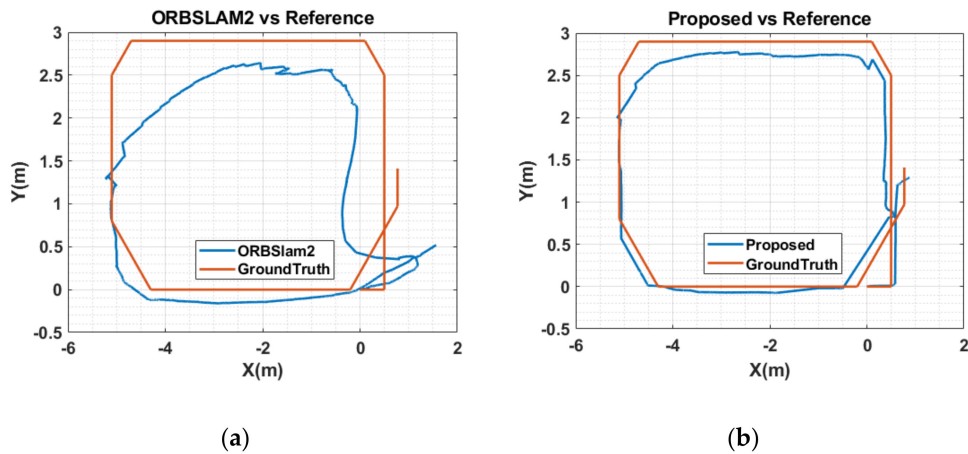

**Figure 8.** Estimated Trajectories for Floor10_3 Sequence: (**a**) ORB-SLAM2, (**b**) SLAM system.

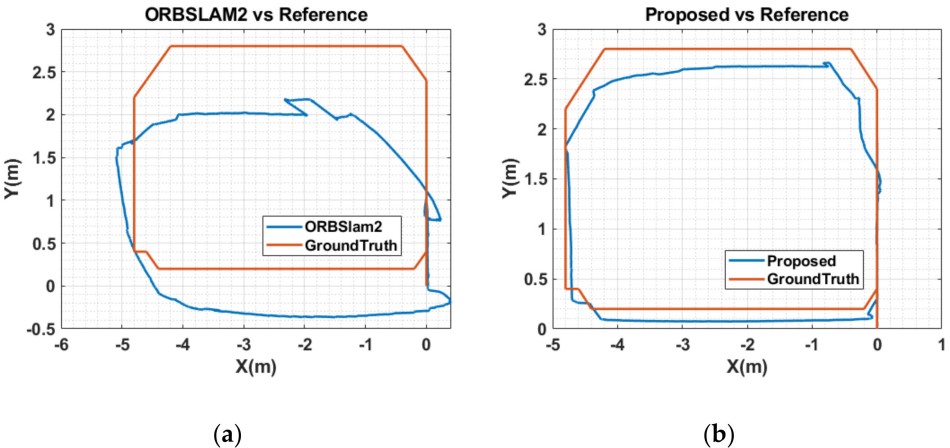

**Figure 9.** Estimated Trajectories for Floor10_4 Sequence: (**a**) ORB-SLAM2, (**b**) SLAM system.

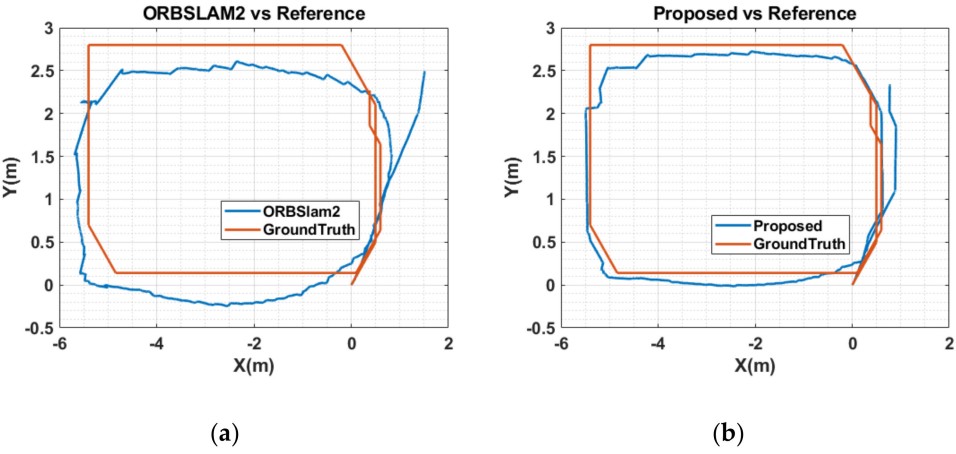

**Figure 10.** Estimated Trajectories for Floor10_5 Sequence: (**a**) ORB-SLAM2, (**b**) SLAM system.

Table 4 shows a comparison between the ORB-SLAM2 and the proposed system. The tests were performed on an Intel Core i7 laptop PC with 16 GB of RAM and an RTX-2070 Nvidia graphics card. ORB-SLAM2 outperforms the proposed solution in both RGB-D and stereo modes; however, with the reported accuracy improvements and the limited speed that characterize most indoor applications, sacrificing the higher frame rate for accuracy could be considered in many cases.

**Table 4.** Frame rate comparison between ORB-SLAM2 vs. Proposed.

|  | **RGB-D** | **Stereo** |
| --- | --- | --- |
| **ORB-SLAM2** | 23 fps | 10 fps |
| **Proposed** | 17 fps | 8 fps |

## 6. Conclusions

The human brain can perform navigation-related tasks flexibly and with very low accumulated errors. In this chapter, an improved visual SLAM system was presented. A new indoor SLAM system is inspired by the human ability to understand and relate semantic and geometrical information and then exploit it to navigate the environment accurately. The proposed system incorporated RoomNet, a CNN network designed to detect geometric properties of the image scenes and successfully used this information to enhance its trajectory estimation. The proposed SLAM solution utilized the geometric constrained between the environment corners to correct the system drifts and create an improved loop closure approach. The presented work included the creation of an object model database to provide more robust object tracking. The proposed system showed robustness and accuracy in all tested sequences and outperformed its backbone SLAM engine, ORB-SLAM2, especially against pure rotations.

**Author Contributions:** Conceptualization, A.M. and M.A.; methodology, A.M.; software, A.M.; validation, A.M. and M.A.; formal analysis, A.M.; investigation, A.M.; resources, A.M.; data curation, A.M.; writing—original draft preparation, A.M.; writing—review and editing, A.M.; visualization, A.M.; supervision, M.A.; project administration, M.A.; funding acquisition, M.A. All authors have read and agreed to the published version of the manuscript.

**Funding:** This research was funded by the Natural Sciences and Engineering Research Council (NSERC) of Canada, Discovery Grant RGPIN-2017-06261.

**Institutional Review Board Statement:** Not applicable.

**Informed Consent Statement:** Not applicable.

**Conflicts of Interest:** The authors declare no conflict of interest.

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
