# Peer review of "Improved Visual SLAM Using Semantic Segmentation and Layout Estimation"

_robotics, doi:10.3390/robotics11050091_

Round 1

Reviewer 1 Report

This paper proposes an improved visual SLAM system using semantic segmentation and layout estimation. The SLAM system is inspired by the human ability to understand the semantic and geometrical information, which is different from traditional methods and has certain innovation. However, the reviewer does not think this paper can be accepted as present version, and some points should be addressed.

(1)The abstract part focuses on describing the mechanism of human brain navigation, however the description of the improvement of visual SLAM is inadequate, which leads to the unclear description of the innovation in this paper.

(2)The description of five stereo/RGBD image sequences generated from the AI2Thor in the experiments part should be provided, as they are not public datasets, and how to get the ground truth needs to be supplemented. Furthermore, the device for the algorithm should be mentioned in the paper.

 (3)The proposed system uses ORBSLAM2 as its SLAM backbone and propagates the semantic and geometric inferences across the tracking, mapping and loop closure tasks, whether the improved system can run online and maintain real-time performance, please make a supplementary explanation. 

Reviewer 2 Report

This work proposed a new indoor SLAM system based on ORB-SLAM2. It is inspired by the human ability to understand and relate semantic and geometrical information to accomplish accurate indoor navigation. Two major improvements are introduced to the proposed system based on ORB-SLAM2. First, in the tracking thread, the proposed SLAM system incorporates trained RoomNet to identify and track the cuboid vertices of the surrounding space to predict room layout keypoints. Second, in the loop closure thread, the proposed SLAM system is based on layout corner detection and is performed by the layout management module in the local mapping. This work also includes the creation of an object model database and validates the proposed SLAM solution on multiple sequences generated from the AI2Thor environment simulator. Evaluation experiments demonstrate that the proposed SLAM system outperforms ORB-SLAM2.

Some comments are as below:

1.     Generally, although the system is a combination of two off-the-shelf methods (ORB-SLAM2 and RoomNet), each thread of the ORB-SLAM2 is redesigned to exploit the semantic and geometric information and achieve an accurate indoor environmental estimation of trajectory and environment map.

2.     However, some components of the proposed SLAM system should be detailed, e.g., the description of 4.1 and 4.3.3 is too sketchy, which makes them hard to understand.

3.     Some recent works in 2.2.2 are missing, e.g.[1][2]

4.     For experiments, how does the ORB-SLAM2 exploit the annotated semantic segmentation provided by AI2THOR? If it does not exploit it, the comparative experiment seems unfair. The details of the experiment should be explained clearly.

5.     There are some format deficiencies:

(a). The figures in this paper are not clear enough, and the space between the subfigure and the text (“type 9” and “type 10”) should be unified in Figure 1.

(b). It is better to explain the symbols (e.g., “CW”) in formulas 1 and 2, and the space between the formula and context should be unified.

(c). A reference in 2.2.2 is corrupt: “Li et al.[119]”.

(d). Some samples of the AI2Thor dataset should be shown.

(e). In the last sentence of the first paragraph of 5.1, all the first letters of “annotated semantic Segmentation” should be lowercase.

[1] Yuan X, Chen S. SAD-SLAM: a visual SLAM based on semantic and depth information[C]//2020 IEEE/RSJ International Conference on Intelligent Robots and Systems (IROS). IEEE, 2020: 4930-4935.

[2] Qiu Y, Wang C, Wang W, et al. Airdos: Dynamic slam benefits from articulated objects[C]//2022 International Conference on Robotics and Automation (ICRA). IEEE, 2022: 8047-8053.

Reviewer 3 Report

This paper research how the human brain accurately navigates and map unknown environments. Compared to traditional Visual Odometry approaches, which suffers from drift, human navigation is much more accurate. This paper uses ML techniques such as semantic segmentation and layout estimation to imitate human abilities to map new environments. The implementation shows tracking errors are reduced when compared to state-of-the-art approaches.

This research explains how human brain functions by a combination of place cells, grid cells, head direction cells, and brain motor sensors. This work further depicts three strategies used by humans to plan a trajectory: allocentric, egocentric and beacon. Finally, it explains how these strategies are combined to perform the whole navigation task.

 Their approach (semantically improved visual SLAM solution) is inspired by human reasoning in solving navigational tasks. This approach builds an accurate joint map for sparse feature models for foreground objects, the environment's geometric bounds and detected sparse feature points to represent the background. This approach has a dependency on RoomNet trained network and ORBSLAM2.

This method relies on the ORBSLAM2 but also has a dependency on offline created model database. My concern is whether it is always possible to model these offline shapes if they are not known a-prioiri or what happens when the shapes in scenarios are arbitrary. Going forward, this approach uses object points instead of background feature points, what happens in the scenarios where you don’t detect enough object points?  Similarly, is the tracking cuboid corner  robust enough to handle all the cases?

The experiments/simulations presented show the tracking error/drift was reduced by a significant factor by using this proposed approach. The results look impressive. My recommendation would be to consider more scenarios where you don’t have a lot of pre-known offline created models in the scene.

Round 2

Reviewer 1 Report

In general, the reviewer think that the authors address my comments, and no futher comments for the authors.

Reviewer 3 Report

The authors have addressed all my concerns. I am OK with this publication.